# Chlorhexidine-Containing Electrospun Polymeric Nanofibers for Dental Applications: An *In Vitro* Study

**DOI:** 10.3390/antibiotics12091414

**Published:** 2023-09-06

**Authors:** Luana Dutra de Carvalho, Bernardo Urbanetto Peres, Ya Shen, Markus Haapasalo, Hazuki Maezono, Adriana P. Manso, Frank Ko, John Jackson, Ricardo M. Carvalho

**Affiliations:** 1Department of Oral Health Sciences, Division of Restorative Dentistry, Faculty of Dentistry, University of British Columbia, 2199 Wesbrook Mall, Vancouver, BC V6T 1Z3, Canada; luanadc@dentistry.ubc.ca (L.D.d.C.); amanso@dentistry.ubc.ca (A.P.M.); 2Department of Oral Biological and Medical Sciences, Division of Biomaterials, Faculty of Dentistry, University of British Columbia, 2199 Wesbrook Mall, Vancouver, BC V6T 1Z3, Canada; buperes@gmail.com (B.U.P.); rickmc@dentistry.ubc.ca (R.M.C.); 3Department of Oral Health Sciences, Division of Endodontics, Faculty of Dentistry, University of British Columbia, 2199 Wesbrook Mall, Vancouver, BC V6T 1Z3, Canada; yashen@dentistry.ubc.ca (Y.S.); markush@dentistry.ubc.ca (M.H.); 4Department of Restorative Dentistry and Endodontology, Graduate School of Dentistry, Osaka Dental University, Osaka 565-0871, Japan; maezono.hazuki.dent@osaka-u.ac.jp; 5Department of Materials Engineering, Faculty of Applied Sciences, University of British Columbia, 309-6350 Stores Road, Vancouver, BC V6T 1Z4, Canada; frank.ko@ubc.ca; 6Faculty of Pharmaceutical Sciences, University of British Columbia, 2405 East Mall, Vancouver, BC V6T 1Z3, Canada

**Keywords:** nanofiber, controlled release preparation, chlorhexidine, anti-bacterial

## Abstract

Chlorhexidine is the most commonly used anti-infective drug in dentistry. To treat infected void areas, a drug-loaded material that swells to fill the void and releases the drug slowly is needed. This study investigated the encapsulation and release of chlorhexidine from cellulose acetate nanofibers for use as an antibacterial treatment for dental bacterial infections by oral bacteria *Streptococcus mutans* and *Enterococcus faecalis*. This study used a commercial electrospinning machine to finely control the manufacture of thin, flexible, chlorhexidine-loaded cellulose acetate nanofiber mats with very-small-diameter fibers (measured using SEM). Water absorption was measured gravimetrically, drug release was analyzed by absorbance at 254 nm, and antibiotic effects were measured by halo analysis in agar. Slow electrospinning at lower voltage (14 kV), short target distance (14 cm), slow traverse and rotation, and syringe injection speeds with controlled humidity and temperature allowed for the manufacture of strong, thin films with evenly cross-meshed, uniform low-diameter nanofibers (640 nm) that were flexible and absorbed over 600% in water. Chlorhexidine was encapsulated efficiently and released in a controlled manner. All formulations killed both bacteria and may be used to fill infected voids by swelling for intimate contact with surfaces and hold the drug in the swollen matrix for effective bacterial killing in dental settings.

## 1. Introduction

Electrospun nanofibers are finding uses as effective drug-delivery biomaterials. The nanoscale properties of these fibers offer the advantage of creating fiber mats with flexibility for good handling and a considerable increase in contact surface area. This allows for improved adhesion to tissues, making these mats attractive for numerous applications in health sciences [1].

Electrospinning is the most popular method for nanofiber production due to the ease of equipment handling and the versatility of the composition of polymeric solutions [2]. The potential application of nanofibers in various fields of dentistry has been investigated, very often as drug release systems, using a wide variety of polymers and drugs [3,4,5,6,7,8]. Souza et al. [8] review anti-infective nanofiber technology that describes all the work conducted with numerous antibiotics, including tetracycline, ciprofloxacin, metronidazole, minocycline and doxycycline, for dental applications. Depending on the polymers employed for nanofiber production, the release can be controlled by different mechanisms (diffusion or diffusion with the degradation of the matrix) [9]. Certain dental applications require an antibiotic delivery system to be highly flexible, swell in aqueous media, and be easily squeezed into voids, such as root canal channels or caries decay sites [8]. For nanofiber systems, this might be achieved with more hydrophilic, cellulose-based polymers electrospun with small-diameter fibers to maximize flexibility and surface area for rapid wetting/swelling. This water absorption would be limited by void space and not create any significant hydrodynamic pressure. 

Chlorhexidine is one of the most commonly prescribed antiseptic agents in dental fields. It has long-lasting antibacterial activity compared to other similar agents, with broad-spectrum action, and it has been shown to reduce plaque, gingival inflammation and bleeding [10,11,12]. Despite a recent systematic review that highlighted the resistance of some microorganisms to chlorhexidine, chlorhexidine rinses favor the resolution of microbial dysbiosis by lowering community diversity and promoting widespread reductions in bacterial genera [13]. Its use is considered a powerful co-adjuvant to mechanical oral hygiene [14]. Several studies have proposed its use as an enzyme inhibitor to reduce collagen degradation [15,16,17,18,19]. Although chlorhexidine may combat collagen degradation and provide benefits to dentin [20], it may present some adverse effects when used in mouthwashes with concentrations higher than 0.2%, which may limit its use in the required extended residence time of the agent in the oral setting [12]. Generally, chlorhexidine application in a cavity is in liquid or gel form, making it difficult to control the amount of agent remaining in the cavity. In these application forms, the chlorhexidine available is restricted to that in contact with the exposed area, and a depot reservoir to supply additional chlorhexidine for long-term action is not presently available. Moreover, any other bioactive agent to be applied would need a separate procedure.

Cellulose acetate is a biocompatible polymer conducive to electrospinning into flexible mats [21,22,23,24,25,26,27]. The material has uses in dentistry as a composite membrane [28] or in the form of electrospun membranes [26,27]. Antibiotic-loaded cellulose acetate electrospun membranes were shown to be effective treatments for diabetic wound treatment [29], underlining the long-term compatibility of the material with sensitive tissues. The objective of this study was to investigate the encapsulation and release of chlorhexidine from cellulose acetate nanofibers for use as an antibacterial treatment for dental bacterial infections by oral bacteria *Streptococcus mutans* and *Enterococcus faecalis*. Because such fibers would likely release the chlorhexidine rapidly, a titanium binding agent was incorporated into the CA fiber mats to slow the release of the drug.

## 2. Results

### 2.1. Morphological Characterization

SEM images showed randomly oriented fibers in all groups. The mean fiber diameter (±standard deviation) of all groups is presented in Table 1.

One-way ANOVA analysis showed that the different formulations of the fibers produced did not result in different fiber diameters (*p* > 0.05). Figure 1, Figure 2, Figure 3, Figure 4 and Figure 5 show nanofibers from the five experimental groups under three magnifications each.

### 2.2. Nanofiber Mat Water Absorption

All nanofiber mats absorbed water rapidly, reaching levels of approximately 600% within 5 min followed by minor increases in water absorption over the next two hours. There was little difference in water absorption between fiber mats, but the chlorhexidine-soaked mats showed slightly higher absorption levels that were significant at the two- and three-hour time points (*p* < 0.05) (Figure 6).

### 2.3. Drug Release

Chlorhexidine release from the different mats was characterized by a burst phase over the first 2 h, followed by a slow release over the rest of the experiment (Figure 7). The mats containing 1.2 wt% chlorhexidine showed statically higher increased release when compared to the other groups in the first 2 h. However, the release from the group where the fibers received the post-spin treatment was higher over time, showing the ability to still release a certain amount of CHX after the three months (last data), which is significantly higher when compared to the other groups (*p* < 0.05).

### 2.4. Antibacterial Assay

The antibacterial assay showed that inhibition halos were formed against *S. mutans* and *E. faecalis* in all mats containing chlorhexidine (Figure 8, Figure 9 and Figure 10). The mats with 1.2 wt% of the drug and the one treated after the spinning process showed statistically higher inhibition than both the positive control (chlorhexidine solution at 300 µg/mL) and the mats with 0.3 wt% of the drug (*p* < 0.05). In general, the inhibition was higher against *S. mutans* than *E. faecalis* for all groups containing CHX. The initial concentration calculated for *S. mutans* was 5.0 × 10^8^ CFU/mL, and for *E. faecalis*, it was 1.2 × 10^8^ CFU/mL. A comparison of the size of the diameter of the halos are represented graphically (Figure 8 and Figure 9) and through images (Figure 10A,B). 

### 2.5. Minimum Bactericidal Concentration

The concentration of chlorhexidine solution to prevent the dental biofilm bacteria growth was detected with solutions at 270 µg/mL and 27 µg/mL. The MBC value was determined to be 2.7 μg/mL against the plaque bacteria.

## 3. Discussion

Cellulose and derivates are polymers often used for nanofiber production through the electrospinning process [21,22]. A great number of publications have suggested the use of cellulose acetate nanofibers for biomedical applications [23,24,25] and often specifically for dental applications [26,27]. Furthermore, depending on the application, it can be blended with other polymers to control drug diffusion and release and is a suitable matrix for the incorporation of antimicrobial compounds [24,25,30,31]. Cellulose acetate has been described as a suitable biomaterial for obtaining a prolonged drug release from cast films [32], suggesting the potential for similar properties from nanofiber mats. The strategy of incorporating a drug into the fibers may be achieved in a number of ways: by evenly distributing or dissolving the active agent in the polymer solution before electrospinning, by confining the active agent in the core of the fiber through coaxial electrospinning, by encapsulating the active agent in nanostructures before dispersing them in the electrospinning solution, by post-treatment of the fiber after electrospinning to convert a precursor to its active form, or the attachment of the active agent onto the fiber surface [33]. In most studies to date, the active agents are blended into the polymer solutions before electrospinning [34]. This technique is simple and able to accommodate a large range of active agents, and the resultant fibers tend to release the active agents in a burst fashion (essential for the initial bacterial killing) followed by a slower release to top up and maintain the bactericidal concentration.

For this investigation, we used a similar method to Chen et al. [35] who encapsulated chlorhexidine into cellulose acetate nanofibers with an average diameter of 950 nm. The use of a commercial electrospinning machine in this study allowed for much finer control of spinning conditions than the lab-assembled systems used by others. We found that using high polymer concentrations in DMF (6%) with the control of chamber humidity and temperature and lower voltage (14 kV), shorter target distance (14 cm), and slower drum rotation with spinning over extended times (12 h) allowed for the production of much smaller-diameter nanofibers in very smooth easy-to-handle sheets. The optimal solution concentration choice was based on a pilot study where the concentrations of 3, 5, and 6 wt% of CA were tested with 0.2 or 0.3 wt% of PEO. The 6 wt% CA and 0.2 wt% PEO were very stable and with suitable viscosity, producing a mat with the best characteristics. It is known that fiber diameter has a direct impact on fiber properties and may eventually influence drug release profiles [34,36]. The task of transforming polymeric solutions into nanofibers is governed by factors related to the process itself, the environment, and factors related to the polymeric solution, such as the choice of solvent, viscosity, and polymer concentration. Although “nano” is normally meant to mean a size range under 100 nm, most of the publications have applied the term “nanofibers” for their fabrics, even in cases where the average diameters were 200–800 nm (submicron), obtained in major works where cellulose acetate fibers were produced by electrospinning, and also in our experiment. The fibers formed were uniform, well distributed, and free of beads, and the mats were easy to handle. The drug incorporation did not interfere with the fiber characteristics. In this study, by using a commercial electrospinning machine, fine control over all spinning parameters was allowed, and very slow overnight spinning was found to give optimal lower-diameter fibers formed into easy-to-handle mats. The manufacturing process was considered a great success, as following many trial-and-error pilot studies, the effective encapsulation of chlorhexidine in usable nanofiber mats was achieved. Furthermore, these mats allowed for the release of the drug and provided excellent antibacterial effects against two particularly problematic dental bacteria. 

The antibacterial action of these nanofiber formulations of chlorhexidine was tested against common oral pathogens, frequently related to active carious lesions or dental infections. Dental biofilm may contain thousands of different bacteria, so it was interesting to assess a bactericidal concentration for a culture of these bacteria because a nanofiber formulation might encounter preexisting biofilm deposits on teeth rather than low-level infection with just one or two bacteria. In our studies, all plaque bacteria were killed using a chlorhexidine concentration of 27 μg/mL. The specific bacteria *S. mutants* and *E. faecalis* are particularly difficult to kill using chlorhexidine with high MBC values, often reported to be close to 100 μg/mL for both bacteria [35,37]. Certainly, then, any drug released from nanofibers should be able to maintain concentrations above this value for some time after placement to achieve the initial first killing of localized infections [8]. In this study, the nanofiber formulations released the drug into release media at levels reaching 5–25 μg/mL over 24 h, but this was into 20 mL release media. These mats measured 5 cm by 1 cm and when moist, this size of film was easily manipulated down into a wet ball so that inside a void, slow drug diffusion from the inside of such a mass might help maintain anti-infective concentrations at the interface with tissues where drug clearance mechanisms would normally reduce drug concentration. A particular limitation of this study is that no animal studies were possible to test the practical application of these nanofiber mats, so these projections of *in vivo* application and effective void concentrations are speculative. 

The nanofiber mats containing either low (0.3 wt%) or high (1.2 wt%) loadings of chlorhexidine were able to effectively eliminate both bacteria in agar plate tests with a similar efficiency to chlorhexidine alone (Figure 8, Figure 9 and Figure 10). 

Since chlorhexidine remains the gold-standard antiseptic in dentistry, being able to act in several dental fields [14], many authors have tried to maintain the chlorhexidine release for long periods through many different strategies, including nanofiber technology [38,39,40,41,42,43,44,45]. In this study, it is assumed that the high levels of drug release after placement in a moist dental void setting would achieve bactericidal concentrations and the compressed swollen mat would retain the released drug in the matrix to maintain the high local concentration. This scenario is totally different from the use of chlorhexidine solution washing of tissue surfaces where the drug is clearly washed away from the target area rapidly. 

This study, despite the limitations related to the use of the electrospinning method, which may generate fiber mats with slightly different characteristics among the samples and the simple method of drug incorporation, successfully provided the manufacture of biocompatible drug-loaded nanofiber mats that might allow for easy dry placement (forceps pushing fibers into void) so that the matrix would absorb enough water, fill the void, release chlorhexidine and ensure immediate contact with potentially infected tissue surfaces. The experiments described in this paper are considered fundamental studies to generate evidence for future research.

## 4. Materials and Methods

### 4.1. Nanofiber Mat Production 

The fibers were produced from a solution containing cellulose acetate Mn ~ 50,000 (CA) and Polyethylene oxide Mw ~ 5,000,000 (PEO) (Sigma-Aldrich, St. Louis, MO, USA), the latter essential to provide the appropriate viscosity of the solution to allow for the formation of the fibers. The polymers were dissolved in N, N-Dimethylformamide (DMF, Fisher Scientific, Waltham, MA, USA), considered a suitable solvent for CA and chlorhexidine powder (CHX). First, the high-molecular-weight PEO (24 mg or 0.2 wt%) was dissolved in a beaker with 12 g of DMF and stirred at 350 rpm at 60 °C for 30 min. After the achievement of a clear solution, 720 mg (6 wt%) of the cellulose acetate powder was added, mixed with a metal spatula and left stirring for 45 min at 60 °C. For the control group 1 (CA-PEO), the clear solution was transferred to a 10 mL plastic syringe to fit in a Nanofiber Electrospinning Unit (NEU–Kato Tech, Kyoto, Japan) to start the electrospinning process. For the groups that received chlorhexidine, titanium triethanolamine was included in the formulation to potentially bind the drug and potentially slow the rate of drug release. A volume of 10 mL of the CA-PEO-DMF solutions was added to 0.1 mL (1 wt%) of Tyzor^®^TE (TTE) (80 wt% titanium triethanolamine in isopropanol), and different amounts of chlorhexidine diacetate were added according to each group (0.3 wt% or 1.2 wt%) and stirred for 10 min at 90 °C. The second control group received TTE but did not receive CHX. The electrospinning parameters were applied voltage of 14 kV; distance between the needle tip and the collector plate of 14 cm; target speed of 1.0 m/min; and traverse speed of 1 cm/min. The syringe pump speed was 0.065 mm/min. Temperature and humidity in the chamber were monitored. Air humidity was kept under 30%, controlled by a dryer unit, and temperature varied between 24 °C and 27 °C. The collector plate was covered with an aluminum foil, the fibers were produced over 12 h in a random mode, and no attempt to align them was made. The formed fiber mats were gently removed from the collector plate and stored in a sealed plastic bag away from heat and humidity until use.

The last experimental group was obtained with a post-spin treatment of the CA-PEO fiber mats with a chlorhexidine solution. The electrospun fiber mats obtained from 6 wt% CA and 0.2 wt% PEO were immersed for 1 h in 10 wt% titanium triethanolamine solution in isopropanol, which was obtained by dilution of the TTE solutions supplied by the manufacturer. The fiber meshes were cured in an oven at 110 °C for 10 min to bind TTE to CA. The fibers were then rinsed with water several times and dried. The resulting fibers were placed in 5% (*w*/*v*) chlorhexidine digluconate aqueous solution for 1 h and cured at 90 °C for 30 min to immobilize the CHX via the titanate linkers. The treated fibers were rinsed with water several times and dried under vacuum to constant weight. 

Based on the composition/treatment of the mats, five experimental groups were formed (Table 2). Triplicate fiber mats were produced for each group.

### 4.2. Nanofiber Water Absorption Studies

Twenty mg sections of each nanofiber mat were placed on Millipore vacuum filter membranes. These mat sections had been pre-weighed both dry and after wetting followed by vacuum removal of excess water. The mats and filter combination were wetted and left for various times in a Petri dish. The filter and mats were removed from water and tilted with forceps to remove excess water and then vacuum treated for three seconds to remove any obvious remaining excess water. The filter and swollen mats were then weighed before returning to the water. The percentage of water absorption was then simply calculated by the ratio of dry weight to wet weight.

### 4.3. Morphological Characterization of Nanofibers

The nanofibers’ morphological characteristics were initially observed under an optical microscope during the electrospinning process (Nikon Eclipse LV100 Optical Microscope, Tokyo, Japan). Samples were collected on a glass slide placed next to the revolving drum to confirm the production of the fibers. A scanning electron microscope (SEM S-238ON, Hitachi, Tokyo, Japan) was used to evaluate the final electrospun mats with an acceleration voltage of 10 kV. The mats were analyzed to evaluate the structure of the fibers. Randomly selected areas of each mat were cut into 5 × 5 mm squares and mounted on stubs with carbon tape (*n* = 3). The stubs were then coated with platinum/palladium (Pt/Pd) with an ion sputter coater (Hitachi E-1030 Ion Sputter Coater, Hitachi, Tokyo, Japan). Random images were taken from the selected pieces of each mat. Average fiber diameter was calculated based on 15 random measurements from pictures taken at the same magnification, using image software (ImageJ version 1.54, NIH, Stapelton, NY, USA).

### 4.4. Chlorhexidine Release Profile

In order to analyze the chlorhexidine released from the fiber mats, samples obtained from each fiber mat (5 × 1 cm) were immersed in vials with 20 mL of distilled water, which were capped and kept in a mixer machine at 100 rpm. The drug release analysis was performed by removing aliquots of the drug-containing solution from each vial, in specific periods (1 h, 2 h, 3 h, 4 h, 5 h, 6 h, 7 h, 8 h, 24 h, 48 h, 7 d, 14 d, 30 d, and 90 d). After each sampling, the media were replaced with 20 mL of freshwater. Chlorhexidine release was analyzed using UV-visible spectrophotometer (UV-1800, Shimadzu, Kyoto, Japan) with the wavelength for chlorhexidine detection at 254 nm. Chlorhexidine release was presented as μg/mL not a percentage of the total, as it was not possible to determine the total amount of encapsulated drug in the formulations.

### 4.5. Antibacterial Assay

*Streptococcus mutans* (NTCC# 10449) and *Enterococcus faecalis* (181) were the dental bacteria used in this study. These bacteria are commonly found in dental biofilms or root canal infections [30]. The preparation of bacterial suspension was conducted with the removal of 20 mL of *S. mutans* from the stock (frozen at −80 °C) of the UBC Endodontics Lab. Bacteria were cultured overnight in 5 mL of brain–heart infusion broth (BHI), at 37 °C in aerobic conditions. After that, 20–50 µL of the overnight culture was put into 8 mL of BHI to obtain another overnight growth solution. On the other hand, the preparation of *E. faecalis* was conducted by the removal of one colony from the stock on BHI agar and placement on a new agar plate in an incubator at 37 °C under aerobic conditions. After the overnight growth, 5–10 colonies were removed from the plate and added to 8 mL of BHI at 37 °C, in aerobic conditions. For the test execution, 500 µL of each diluted bacterial solution (100×) was plated on the new BHI agar plate and left to dry at room temperature. Discs with a 6 mm diameter of the fiber mats were positioned over the smeared bacteria and incubated overnight at 37 °C. Filter paper with water and the CA mats with no drugs were used as the negative control, and filter paper immersed in chlorhexidine solution (concentration 300 µg/mL) was used as the positive control. The fiber mats with different concentrations of chlorhexidine had their antibacterial action tested based on the inhibition halos formed around the discs. After the incubation period, inhibition halos were measured. The experiments were performed in triplicate.

### 4.6. Minimum Bactericidal Concentration

To determine the minimum bactericidal concentration of chlorhexidine solution (the concentration that reduces bacterial numbers to below an assay detection limit) on a mixed plaque sample, the bacterial suspension was prepared (mix of human plaque plus BHI) with an optical density of 0.1 at 405 nm for a colony-forming unit concentration of approximately 1 × 10^7^ cfu/mL. This was diluted to 1 × 10^5^ cfu and mixed with chlorhexidine at 270, 27, 2.7, 0.27, and 0.027 μg/mL (final concentration). Then, 20 μL of each solution was plated on BHI agar plate (triplicates) to be incubated in aerobic conditions at 37 °C for three days. The minimum concentration of chlorhexidine to prevent the multispecies bacterial growth was determined in the plates where no bacterial colonies were observed.

### 4.7. Statistical Analysis

Differences in the average fiber diameters, drug release at each period, and inhibition halo diameters were compared using one-way ANOVA for each experiment (α ≤ 0.05), with the normality test conducted using Shapiro–Wilk test and post hoc Tukey’s test to detect significant pairwise comparisons between groups (*p* ≤ 0.05). Sigma Plot software version 14.5 (Systat Software, San Jose, CA, USA) was used for statistical analysis.

## 5. Conclusions

The chlorhexidine-loaded cellulose acetate nanofiber mats developed in this study may offer dentists an improved treatment method over chlorhexidine rinsing of surfaces. Chlorhexidine was encapsulated efficiently and released in a controlled manner. The mats were successful in eliminating *S. mutans* and *E. faecalis*, which are two particularly difficult-to-kill dental bacteria and would swell into a soft gel-like matrix that might enable intimate contact with dental surface for extended periods of time.

## Figures and Tables

**Figure 1 antibiotics-12-01414-f001:**
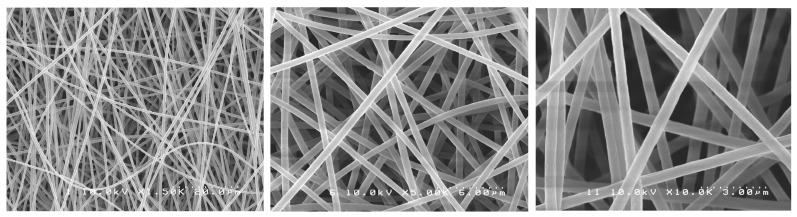
SEM photomicrographs of CA-PEO mats under different magnifications.

**Figure 2 antibiotics-12-01414-f002:**
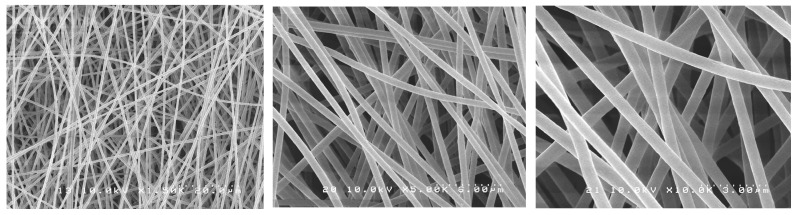
SEM photomicrographs of CA-TTE mats under different magnifications.

**Figure 3 antibiotics-12-01414-f003:**
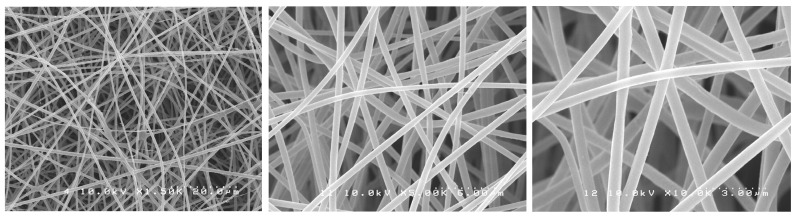
SEM photomicrographs of CA-CHX 0.3 mats under different magnifications.

**Figure 4 antibiotics-12-01414-f004:**
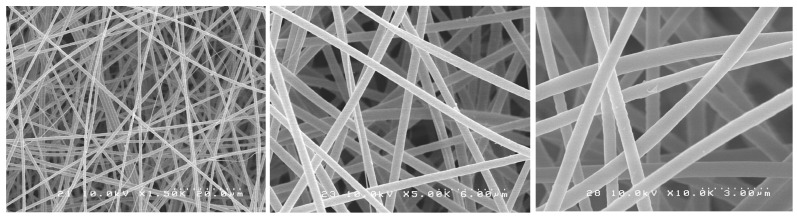
SEM photomicrographs of CA-CHX 1.2 mats under different magnifications.

**Figure 5 antibiotics-12-01414-f005:**
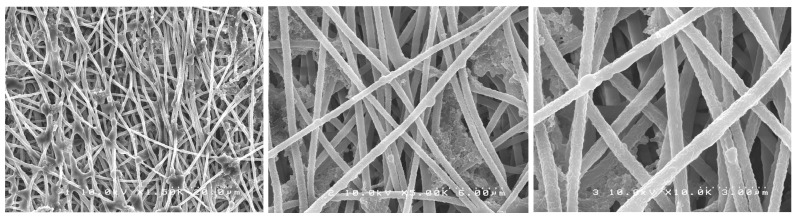
SEM photomicrographs of CA-PEO with post-spin treatment mats under different magnifications.

**Figure 6 antibiotics-12-01414-f006:**
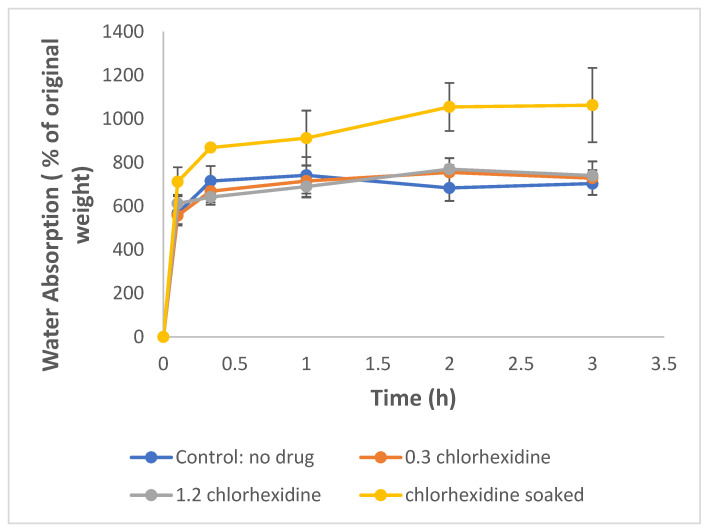
Water absorption of chlorhexidine-loaded cellulose acetate nanofiber mats (0.3 and 1.2 indicate the loading % of chlorhexidine in cellulose acetate).

**Figure 7 antibiotics-12-01414-f007:**
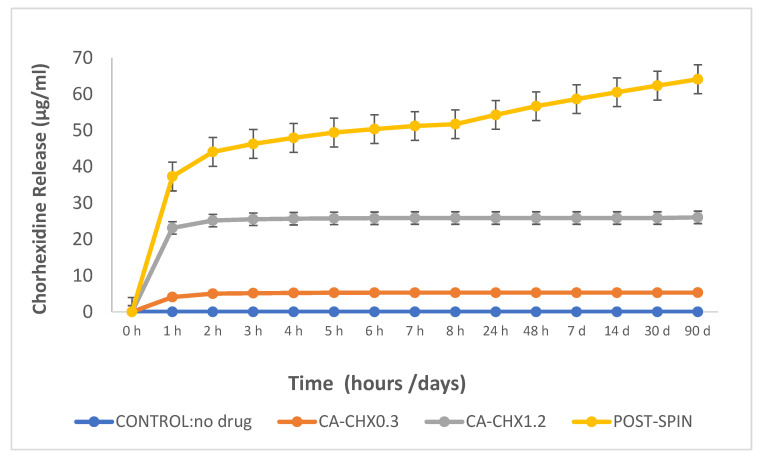
Release of chlorhexidine (μg/mL) from the nanofiber mats.

**Figure 8 antibiotics-12-01414-f008:**
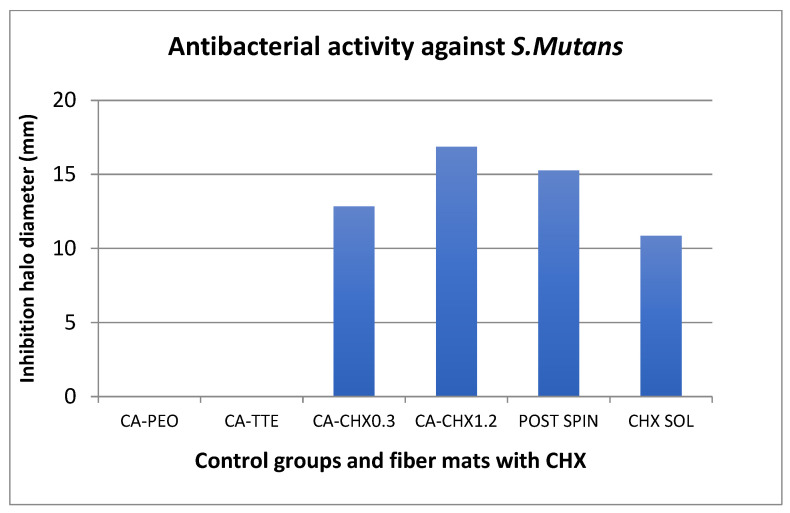
Comparison of the size of the inhibition halos against *S. mutans* formed around the different fiber mats and controls.

**Figure 9 antibiotics-12-01414-f009:**
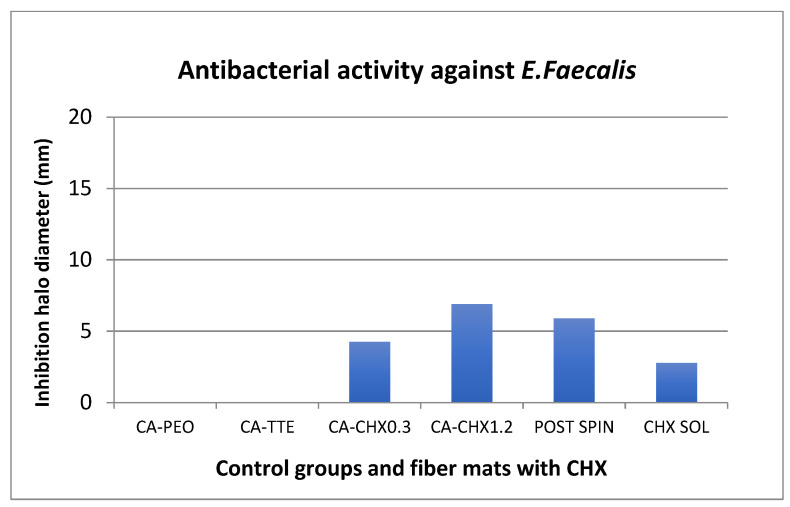
Comparison of the size of the inhibition halos against *E. faecalis* formed around the different fiber mats and controls.

**Figure 10 antibiotics-12-01414-f010:**
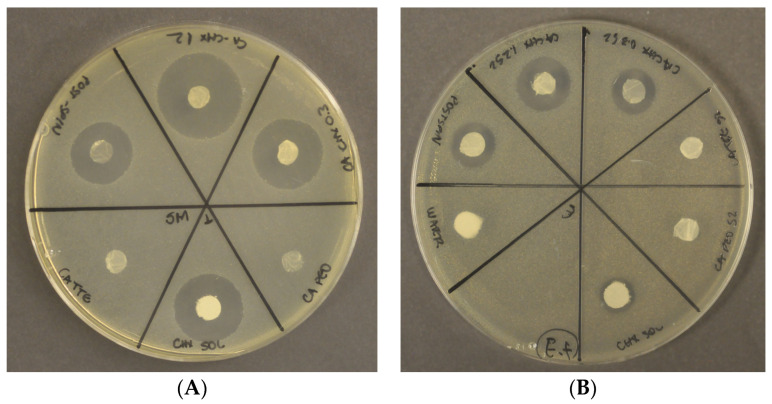
The inhibition halos observed when the experimental mats were applied against *S. mutans* (**A**) and *E. faecalis* (**B**).

**Table 1 antibiotics-12-01414-t001:** Mean fiber diameter (nm) present in the different mats.

CA-PEO	CA-TTE	CA-CHX 0.3	CA-CHX 1.2	CA-PEO POST-SPIN
588 (±57)	600 (±54)	569 (±76)	613 (±122)	584 (±62)

CA-PEO: Cellulose acetate and Polyethylene oxide; CA-TTE: Cellulose acetate, Polyethylene oxide and Titanium triethanolamine linker; CA-CHX 0.3: Cellulose acetate, Polyethylene oxide, Titanium triethanolamine linker and Chlorhexidine diacetate powder at wt% 0.3; CA-CHX 1.2: Cellulose acetate, Polyethylene oxide, Titanium triethanolamine linker and Chlorhexidine diacetate powder at wt% 1.2; CA-PEO POST-SPIN: Cellulose acetate and Polyethylene oxide immersed in 5% (*w*/*v*) Chlorhexidine digluconate aqueous solution binded via the Titanium triethanolamine.

**Table 2 antibiotics-12-01414-t002:** Polymers, solvents, and drugs added to the solutions of the experimental groups.

Composition					
Groups	CA	PEO	TTE	DMF	CHX
CA-PEO	0.78 g	0.025 g	-	12 g	-
CA-PEO-TTE	0.78 g	0.025 g	0.12 g	12 g	-
CA-PEO-TTE-CHX 0.3	0.78 g	0.025 g	0.12 g	12 g	0.038 g
CA-PEO-TTE-CHX 1.2	0.78 g	0.026 g	0.13 g	12 g	0.15 g
CA-PEO-POST-SPIN	0.78 g	0.025 g	-	12 g	Post-spin treatment

CA: Cellulose acetate; PEO: Polyethylene oxide; TTE: Titanium triethanolamine linker; DMF: N, N-Dimethylformamide; CHX: Chlorhexidine diacetate powder.

## Data Availability

Not applicable.

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
