# Peer review of "Chlorhexidine-Containing Electrospun Polymeric Nanofibers for Dental Applications: An In Vitro Study"

_antibiotics, 2023, doi:10.3390/antibiotics12091414_

Round 1

Reviewer 1 Report

“Chlorhexidine-containing electrospun polymeric nanofibers for dental applications” was submitted to Antibiotics.

This study investigated the encapsulation and release of chlorhexidine from cellulose acetate nanofibers for use as an antibacterial treatment for dental infections. The authors concluded that all drug-loaded nanofiber mats effectively killed bacteria and may be used to fill infected voids.

The manuscript deals with an interesting issue; however, there are several concerns related to the study.

Title: It should be noted that it is an in vitro study.

Abstract

- The objective does not coincide with the one presented in the text of the introduction. The latter is more clearly described.

- Please consider writing the microorganisms in the proper way. Write them in italics and completely the first time they are mentioned. Please consider this recommendation throughout the manuscript.

- Considering that the objectives describe two main outcome variables (encapsulation and release of chlorhexidine), the authors should present conclusions in this regard.

Keywords: They must be reviewed. Some are not MeSH terms.

Introduction

- “Certain dental applications….”. Add references to this paragraph. In this paragraph, the authors also mention “antibiotic delivery system”. Does this also apply to antiseptics? Please revise.

- “Chlorhexidine is one of the most commonly…...”. Recent systematic scoping reviews have highlighted the resistance of some microorganisms to chlorhexidine. Also, other disadvantages of this antiseptic must be mentioned.

Please consider the section structure recommended in the instructions to the authors.

M&M

Please indicate the statistical test used to determine the normal distribution of the data with its respective p-value.

Results

Please define MBC.

Discussion

- “In most studies to date, the active agents…..”.  Please add references to this paragraph.

- “It is known that fiber diameter has a direct impact…..”. Please add references to this paragraph.

- “Certainly, then any drug released from nanofibers…”. Please add references to this sentence.

- “These mats measured 5 cm by 1 cm and were….”. Please add references to this sentence.

- Study limitations should be presented.

Conclusions.

Considering that the bacteria studied are not usually observed in periodontal diseases, caution should be exercised in commenting on it.

moderate editing

Author Response

Please find attached a point by point response to each of the review comments.

Thanks

Reviewer 2 Report

  The paper shows the preparation of cellulose acetate nanofibrous mat releasing chlorhexidine. Specific bacteria for dental application were used. The paper is well-written but has many low-quality images.   Specific comments: Abstract page 1 Slow electrospinning Note: What do you mean by slow spinning, and why is it important?   page 1 The potential application of nanofibers in various fields of dentistry has been investigated, very often as drug release systems, using a wide variety of polymers and drugs.[3–8] Note: Can you please refer to what other groups were doing in this matter? Did anyone use chlorhexidine in nanofibers?   page 2 swell in aqueous media and to be easily squeezed into voids such as root canal channels Note: Would swelling create problems for the patient after a tight squeeze in the root canal channel? 600% increase in mass means there is also a significant volume increase.   Materials and methods page 2 0.065mm/min Note: Can you please provide a volumetric flow rate?   page 3 The fiber meshes were cured in an oven at 110°C for 10 minutes to bind TTE to CA Note: It needs to be better written about why you use TTE.   page 3 Nanofibers Swelling Studies Note: You are not measuring the swelling of the material but its water absorption (uptake). Swelling should be denoted to the measure of fiber diameter increase etc.   page 4 Chlorhexidine release is presented as ug/mL not % of total as it was not possible to determine the total amount of encapsulated drug in the formulations. Note: Why was it impossible to determine the total amount of the drug? Assuming that the solvent evaporated, finding the % of antibiotics in the formulation should be possible.   page 6 Swelling o Note: This is water uptake, not swelling     page 7 Release of chlorhexidine (ug/mL) from the nanofiber mats. . Note: Time points should be presented instead of lines. Moreover, samples not having CHX could be excluded from the graph. Moreover, the chart's title should be removed, and the Y axis should be described for identification if it is a cumulative release, etc. Moreover, This figure does not have a frame, unlike the previous one.   page8 Comparison of the size of the inhibition halos against S. mutans formed around the different fiber mats and controls. Note: Please provide 2d charts and not pseudo 3d that are confusing.     page 9 suggesting the potential for similar properties from nanofiber mats. Note: Not necessarily. Fibrous nanomaterials show more often burst release due to their small size.     page 9 For this investigation we used a similar method to Chen et al [31] who encapsulated chlorhexidine into cellulose acetate nanofibers with an average diameter of 950nm. The use of a commercial electrospinning machine in this study allowed for much finer control of spinning conditions than the lab-assembled systems used by others. Note: But does it make it more scientifically interesting?   page 10 When moist this size of film was easily manipulated down into a wet ball of perhaps 50ul dimension (similar volume to a root canal channel) so that theoretically this size of mat would release drug and achieve local concentrations far in excess of 100ug/ml. Note: 5x1 cm mat can't be manipulated to make a 50 uL diameter ball. You made perfect sink conditions using 20 mL of acceptor medium, but it doesn't mean a higher concentration could be achieved in the root canal. There is vasculature around, and it would be hard to achieve it. Moreover, the release characteristics would definitely change after post-processing fibers into a ball.

English is correct.

Author Response

Please find attached a point by point response to all the review comments

Thanks 

Round 2

Reviewer 1 Report

Some aspects remained unresolved:

Title: It should be noted that it is an in vitro study.

Abstract

- Considering that the objectives describe two main outcome variables (encapsulation and release of chlorhexidine), the authors should present conclusions in this regard.

Keywords: They must be reviewed. Some are not MeSH terms.

Introduction

- “Chlorhexidine is one of the most commonly…...”. Recent systematic scoping reviews have highlighted the resistance of some microorganisms to chlorhexidine.

Please consider the section structure recommended in the instructions to the authors.

M&M

Please indicate the statistical test used to determine the normal distribution of the data with its respective p-value.

Discussion

- “Certainly, then any drug released from nanofibers…”. Please add references to this sentence.

- This study has more limitations that were not presented.

Conclusion

Considering that the objectives describe two main outcome variables (encapsulation and release of chlorhexidine), the authors should present conclusions in this regard.

The manuscript is full of typos; therefore it must be extensively reviewed.

Extensive editing

Author Response

Many thanks for the comments.  We have corrected the document as requested

A point by point reply is included along with the revised document.   Please see attachment. 
